# The Complex Relation between Executive Functions and Language in Preschoolers with Developmental Language Disorders

**DOI:** 10.3390/ijerph17051772

**Published:** 2020-03-09

**Authors:** Andrea Marini, Barbara Piccolo, Livia Taverna, Moira Berginc, Martina Ozbič

**Affiliations:** 1Department of Languages, Literatures, Communication, Education and Society, University of Udine, Udine 33100, Italy; 2Scientific Institute IRCCS “Eugenio Medea”, San Vito al Tagliamento, Pordenone 33078, Italy; martinaozbic@siol.net; 3Struttura Complessa Neuropsichiatria Infantile, Azienda Sanitaria Universitaria Giuliano Isontina, Trieste 34139, Italy; barbara.piccolo@asugi.sanita.fvg.it; 4Faculty of Education, Free University of Bolzano-Bozen, Bolzano 39100, Italy; livia.taverna@unibz.it; 5Ambulatorio per il trattamento riabilitativo della prima infanzia, Casa della sanità di Capodistria/Zdravsteni Dom Koper, Koper 6000, Slovenia; moira.berginc@zd-koper.si

**Keywords:** developmental language disorders, executive functions, phonological working memory, developmental neuropsychology, neurolinguistics

## Abstract

*Backgrounds*: The relationship between linguistic difficulties and cognitive impairments in children with developmental language disorders (DLDs) is receiving growing interest in international research. Executive functions (EF) appear to be weak in these children. The current investigation aims at exploring the relationship between difficulties in two components of EF (i.e., updating and inhibition) and the linguistic and narrative skills of 16 DLD preschoolers matched with 24 typically developing peers. *Methods*: Updating skills were tested by administering the forward and backward digit recall subtests of the Wechsler Scales, while children’s inhibition abilities were assessed by completion of Developmental Neuropsychological Assessment (NEPSY-II) inhibition tasks. Information on the linguistic skills of the participants was collected through a set of subtests included in the Batteria per la Valutazione del Linguaggio in bambini dai 4 ai 12 anni (Batteria per la Valutazione del Linguaggio; BVL_4-12), assessing articulatory and phonological discrimination skills, lexical production/comprehension, grammatical production/comprehension, and narrative production skills. *Results*: Findings revealed that DLD children performed significantly lower than their peers on both updating and inhibitory tasks. Linguistic difficulties were found in the DLD group on articulatory/phonological skills, grammatical production/comprehension, and lexical informativeness on narrative production. Measures of EF correlated with linguistic and narrative measures. *Conclusion*: The current study confirms a significant association between DLD’s performances on EF and displayed linguistic skills, suggesting the need to include the assessment of executive functions to target early intervention rehabilitation programs for children with DLDs.

## 1. Introduction

Growing interest is mounting on the cognitive characteristics of children with developmental language disorders (DLDs). This diagnostic label has been recently introduced to replace previous ones such as specific or primary language impairments. It refers to children who have language disorders not associated with biomedical conditions such as brain injury, neurodegenerative diseases, cerebral palsy, or other conditions linked to genetic or neurological causes [1]. Crucially, it is now well-accepted that such linguistic difficulties may co-occur with a variety of other cognitive impairments affecting, for example, procedural memory [2], motor control [3], and executive functions (EFs; [4,5,6]). Because of their relevant role in cognitive and linguistic development, the study of EFs in DLDs may be particularly important for an accurate interpretation of the communicative and language difficulties observed in these children.

According to Miyake and colleagues [7], EFs are set of cognitive skills (moderately correlated components in a confirmatory factor analysis) necessary to (1) update the ongoing information in working memory while monitoring the execution of a task (updating), (2) inhibit unnecessary or non-pertinent pieces of information (inhibition), and (3) efficiently shift between mental sets (shifting). More recently, Miyake and Friedman [8] proposed a unity/diversity framework for the organization of EFs. According to their model, these three components would share common underlying processes (unity, i.e., what these components have in common) but might depend on component-specific processes (diversity). For example, the ability to maintain the focus on the aim and execution of a task might significantly rely on the joint contributions of the inhibitory (e.g., allowing a speaker to inhibit the production of irrelevant or tangential information while producing a discourse), shifting (needed to flexibly shift between topics and/or perspectives), and updating (implicated in the ability to link different pieces of information exchanged during a conversation) components of EFs. Garon and colleagues [9,10] suggest that such components develop hierarchically in children. Namely, the development of EFs may take place only after the development of sustained attention, i.e., the ability to focus on a specific target for prolonged time periods. As a second step, children would begin to develop the ability to keep the information active in their working memory (i.e., updating). Afterwards, inhibition and eventually shifting would begin to develop. Crucially, later abilities appear to rely on previously acquired skills and may significantly affect complex behaviors such as those required in efficient, communicative exchanges. From this point of view, a task requiring children to shift between mental sets should require the ability to inhibit irrelevant stimuli, keep the relevant ones in working memory, and focus on the task as long as it is needed. This also means that difficulties in the development of earlier skills (e.g., attention skills) might negatively impact on the development of later abilities, such as working memory, inhibition, and/or shifting.

Growing evidence suggests that children with DLDs may have difficulties in maintaining sustained attention on verbal [11,12,13] and non-verbal tasks [14,15,16] in phonological working memory, which is crucial for the updating component of EFs [16,17], and on tasks assessing inhibitory control and/or cognitive flexibility [4]. This apparently also applies to preschoolers with DLDs (e.g., [18,19,20]) and supports the possibility of a domain-general executive function deficit in these children. An interesting issue regards the possible effects of difficulties in EFs on the linguistic and communicative abilities of children with DLD. Message production and comprehension are likely the result of complex multistage processes (e.g., [21,22]) where the three executive components might play relevant roles. Indeed, according to Miyake and colleagues [7] and Mozeiko et al. [23], *shifting* may be involved in the generation of complete episodes within a narrative discourse, in the selection of informative words, and in the ability to monitor the communicative flow; *updating* may be required to generate and understand sentences as well as recall former episodes or episodic contents for an accurate organization of a story; *inhibition* may be critical for monitoring the production of extraneous comments and derailments while telling a story and for the ability to inhibit the semantic competitors while producing or understanding words. The current investigation aims at exploring the potential relation between difficulties in two of the executive components described in Miyake and colleagues [7] (i.e., updating and inhibition) and the linguistic and narrative skills of a cohort of preschoolers with a diagnosis of DLD. Namely, a group of preschoolers with DLDs and one of children with typical development were administered tasks assessing their abilities to (1) keep verbal information in their phonological short-term and working memory while completing a task (updating), and (2) inhibit the production of prepotent responses under specific conditions. Furthermore, they also received tasks assessing their linguistic and narrative abilities. We hypothesize that (1) the participants with DLD would perform worse than controls on tasks assessing inhibition and updating, (2) they would have significant difficulties on tasks assessing production and comprehension, and (3) their performance on the tasks assessing inhibition and updating would be related to their linguistic and narrative difficulties.

## 2. Materials and Methods

### 2.1. Participants Characteristics

Forty monolingual Italian-speaking children were included in the study. They formed an experimental and a control group. The experimental group was formed by sixteen children with a diagnosis of developmental language disorder (DLD) recruited in rehabilitation centers in Italy. The control group was made of twenty-four children with typical development (TD) recruited in mainstream schools. None of the participants had intellectual disability, brain lesions, or auditory difficulties. After Bonferroni’s correction for multiple comparisons (0.05/4 = 0.013), the two groups did not differ on age (t_(38)_ = 1.811; *p* = 0.078; d = 0.606), level of formal education (all of them were attending to the 3rd year of preschool), gender (X^2^ (1, *N* = 40) = 0.233, *p* = 0.630; odds ratio: 0.556), or Raven’s Matrices (t_(38)_ = 2.444; *p* = 0.019; d = 0.782) (see Table 1).

The parents of the participants released their written and informed consent to the participation of their children in the study and to data processing. The study was approved by the Ethical Committee of the Region Friuli Venezia Giulia (CEUR), Italy (protocol n. 23826).

### 2.2. Assessment of Updating Skills

In order to assess their updating skills, all participants were administered the forward and backward digit recall subtests of the Wechsler Scales [24]. The former is a simple span task aimed at evaluating a child’s phonological short-term memory. The latter is a complex span task used to explore the ability of the child to manipulate information in working memory (i.e., it is a task assessing phonological working memory). In the forward digit recall test, children were asked to repeat sequences of digits in the correct serial order. The sequences ranged from 1 to 9 digits that the examiner produced at the rate of 1 digit per second. The number of sequences that the child was able to repeat forms the Forward Digit Recall score. In the backward digit recall test, children were asked to repeat the sequence of spoken digits in reverse order. The number of sequences that the child was able to adequately invert and repeat forms the Backward Digit Recall score. As composite measures derived by span tasks show better test–retest reliability and greater stability for subject classification than do the use of any single measure [25], a Composite Digit Recall score was calculated by summing the Forward and Backward Digit Recall scores.

### 2.3. Assessment of Inhibitory Skills

The inhibition task of the Italian version of the Developmental Neuropsychological Assessment (NEPSY-II, [26]) was used to assess monitoring, self-regulation, and the ability to inhibit automatic responses in favor of novel ones (i.e., prepotent response inhibition; [27]). Namely, this timed test requires the child to look at a series of black and white shapes (i.e., squares and circles) or arrows and name either the shape, direction, or an alternate response (in the inhibition condition). For the purpose of this study, we administered Parts A and B of the test: Part_A (naming condition) requires participants to name the shape of squares and circles or the up or down direction of arrows; Part_B (inhibition condition) requires participants to provide the opposite naming response on the same stimuli (e.g., if the child is shown a circle, (s)he must say “square”; if (s)he sees a square, (s)he must say “circle”).

### 2.4. Procedures of Linguistic Assessment

The linguistic skills of the participants were assessed by administering a selection of the tasks included in BVL_4-12 [28].

### 2.5. Assessment of Articulatory and Phonological Discrimination Skills

Articulatory and phonological discrimination skills were assessed by administering the Naming and Phonological Discrimination subtests of the Batteria per la Valutazione del Linguaggio in Bambini dai 4 ai 12 Anni (Battery for the assessment of language in children aged 4 to 12 (BVL_4-12); [28]). In the naming subtest of the BVL_4-12, children were asked to name up to 77 visual stimuli (drawings) referring to words with different (a) articulatory characteristics, (b) frequencies in Italian (words with low, medium, and high frequency), (c) grammatical class (verbs and nouns), and (d) semantic features (action verbs and nouns pertaining to a number of semantic categories). This test allowed us to have Articulatory and Naming scores. Children providing the right answer and articulating it properly received 1 point for naming and 2 points for articulation. If the correct word was mispronounced, the child scored 1 point for Naming but then was asked to repeat the word that this time was provided by the examiner. In case of correct repetition, the child’s answer was assigned 1 point for articulation. If (s)he still could not utter it correctly, then the child was given 0 points for articulation. If the child did not provide any word for the target picture and could not repeat it properly after the examiner, then he or she received 0 points for naming and 0 points for articulation (maximum naming score = 77; maximum articulation score = 154).

In the phonological discrimination subtest of the BVL_4-12, children listened to a total of 30 couples of words that could be either minimal pairs (*N* = 20) (i.e., those words that differ for just one phoneme, such as in *cane*–*pane*) or identical items (*N* = 10). A percentage of phonological discrimination was calculated subtracting the % of false alarms to the % of correct answers.

### 2.6. Lexical Assessment

Lexical production skills were assessed with the Naming score obtained with the administration of the Naming task of the BVL_4-12. Lexical comprehension was assessed by administering the Lexical Comprehension subtest of the BVL_4-12, where participants were asked to identify which, among four pictures, represented the meaning of the word uttered by the examiner. The four pictures represented a target stimulus, a semantic, a phonological, or an unrelated distracter. For each correct answer, the child received 1 point (maximum Lexical Comprehension score = 18).

### 2.7. Grammatical Assessment

Grammatical production and comprehension skills were assessed with the percentage of complete sentences derived from the multilevel analysis of the discourse production task and the Grammatical Comprehension task of the BVL_4-12, respectively.

In the discourse production task, children were asked to produce a sample of narrative discourse describing one cartoon-story made of six pictures (the ”Nest Story” by [29]). Each storytelling was tape-recorded and transcribed verbatim. The transcriptions underwent a comprehensive, multilevel analysis of discourse production, which focused on both micro- and macrolinguistic aspects of the narrative production [30]. For the purposes of this study, we will focus on the % of complete sentences and on two macrolinguistic measures (i.e., % errors of global coherence and % lexical informativeness, respectively). For a description of these measures, please refer to the Narrative Assessment section). The scoring procedure was performed independently by two raters and then compared. Acceptable inter-rater reliability was set at Cohen’s k ≥0.80. Any residual difference was resolved through discussion. As for the analysis of the participants’ grammatical production skills, a percentage of complete sentences was calculated by dividing the number of grammatical sentences by the number of utterances uttered by each participants applying the following formula: (complete sentences/utterances)*100. For each story description, the total number of utterances was assessed following the criteria illustrated in [30]. These included an acoustic, a semantic, a grammatical, and a phonological criterion. According to the acoustic criterion, an utterance was considered as an emission of sounds delimited by pauses. According to the semantic criterion, an utterance was a conceptually homogeneous proposition. Following the phonological criterion, if a word was interrupted (i.e., there was a false start), then the utterance was considered abruptly interrupted as well. Finally, according to the grammatical criterion, a chain of words formed an utterance if, in absence of long pauses (acoustic criterion), propositional violations (semantic criterion), or false starts (phonological criterion), it formed a grammatically complete sentence (eventually including also subordinate clauses). A sentence was considered grammatically complete if all the arguments required by the verb were inserted correctly in the body of the sentence and if there were no omissions or substitutions of free or bound morphemes.

In the Grammatical Comprehension task, children were asked to identify which, among four pictures, represented the meaning of a sentence uttered by the speaker. The four pictures represented a target stimulus and three morphosyntactic distracters. One point was assigned for each correct answer (maximum Grammatical Comprehension score = 40).

### 2.8. Narrative Assessment

The narratives produced by the participants were analyzed with a measure of discourse organization and one of informative content (i.e., % errors of global coherence and % lexical informativeness, respectively).

Errors of global coherence included the production of utterances that may be tangential, incongruent with the story, propositional repetitions, or simple fillers (for a thorough description of such errors, please refer to [30] and [31]). An utterance was considered tangential when it contained a derailment in the flow of discourse with respect to the information already provided in a preceding utterance. It was considered conceptually incongruent when it included ideas not directly addressed by the stimulus. A propositional repetition was considered a sequence where the subject repeated ideas that had already been provided. Finally, a filler utterance was scored whenever s(he) produced an utterance that was not providing any additional information. A percentage of global coherence errors was calculated with the following formula: (global coherence errors/utterances)*100.

The communicative efficacy of each narrative was measured with a measure of lexical informativeness (i.e., the ability to select and produce words that are morphologically, semantically, and pragmatically appropriate to describe the gist of the story [32]). Namely, such lexical efficacy was determined by considering the total amount of Lexical Information Units (LIUs) produced in a narrative description. The words that had been scored as errors of any kind, as well as words embedded in filler, repeated, incongruent or tangential utterances, were excluded from the count of the lexical information units. The percentage of lexical informativeness was calculated with this formula: (lexical information units/words)*100.

## 3. Results

### 3.1. Assessment of Updating Skills (Phonological Short-Term and Working Memory)

Levene’s test for equality of variances showed that the assumption of homogeneity of variance had not been violated for tasks assessing digit recall. Therefore, potential group-related differences were explored by performing three independent-samples *t*-tests for forward digit recall, backward digit recall, and composite score of digit recall. Statistical significance was set at *p* < 0.017 (0.05/3 dependent variables) after Bonferroni correction for multiple comparisons. These analyses showed that the participants with DLDs had lower scores on all three tasks: forward digit recall (t_(38)_ = 2.653; *p* < 0.012; d = 0.828), backward digit recall (t_(38)_ = 3.188; *p* < 0.003; d = 1.012), composite digit recall (t_(38)_ = 3.754; *p* < 0.001; d = 1.182) (see Table 2).

### 3.2. Assessment of Inhibitory Skills

The ability of the two groups to monitor their performance and inhibit inappropriate responses was explored by administering the inhibition task (Parts A and B) of the NEPSY-II [32]. For these two tasks, scalar scores were considered. The group-related difference was explored by performing two Pearsons’ chi-squares. The alpha level was set at *p* < 0.025 after a Bonferroni correction for multiple comparisons (05/2 = 0.025). These analyses showed that participants with DLDs produced significantly more errors than controls, namely, they obtained lower scalar scores (Part_A: (X_2_ (_3, N=40)_ =13.415, *p* < 0.004; odds ratio: 0.733 (These two odds ratios have been calculated considering how many participants in each group reached at least the 50th centile (i.e., groups 51–75 and >75))); Part_B: (X^2^
_(5, N=40)_ =16.679, *p* < 0.005; odds ratio: 0.111)) on both versions of the task (see Table 3).

### 3.3. Analysis of Phonological/Articulatory Skills

Data regarding articulation and phonological discrimination skills of the two groups of participants are presented in Table 4, where the standardized differences (mean of *z*-scores) between children with DLD and the normative data available for each linguistic task of the BVL_4-12 [28] are presented. Levene’s test for equality of variances showed that the assumption of homogeneity of variance had been violated for both articulation (*p* < 0.001) and phonological discrimination (*p* < 0.005). For this reason, potential group-related differences were explored with two non-parametric Mann–Whitney tests. Statistical significance was set at *p* < 0.025 (0.05/2 dependent variables) after Bonferroni correction for multiple comparisons. These analyses showed that children with DLDs performed worse than children with TD on both articulation (U = 43.00, *p* < 0.001; d = 1.713) and phonological discrimination (U = 65.50, *p* < 0.001; d = 1.325).

### 3.4. Analysis of Lexical Skills

Data regarding the lexical production and comprehension skills of the two groups of participants are presented in Table 5. Levene’s test for equality of variances showed that the assumption of homogeneity of variance had not been violated for either variable. For this reason, potential group-related differences were explored with two independent-sample *t*-tests. Statistical significance was set at *p* < 0.025 (0.05/2 dependent variables) after Bonferroni correction for multiple comparisons. These analyses showed the absence of any group-related difference in either naming (t_(38)_ = 1.713; *p* = 0.095; d = 0.054) or lexical comprehension (t_(38)_ = 1.295; *p* = 0.203; d = 0.403).

### 3.5. Analysis of Grammatical Skills

Data regarding the grammatical production and comprehension skills of the two groups of participants are presented in Table 6. Levene’s test for equality of variances showed that the assumption of homogeneity of variance had been violated only for grammatical comprehension (*p* < 0.026). For this reason, potential group-related differences were explored with one independent-sample *t*-test for % complete sentences and one non-parametric Mann–Whitney test for grammatical comprehension. Statistical significance was set at *p* < 0.025 (0.05/2 dependent variables) after Bonferroni correction for multiple comparisons. These analyses showed that participants with DLD produced fewer grammatically complete sentences than controls on the narrative production task (t_(38)_ = 2.591; *p* < 0.014; d = 0.083) and understood fewer sentences on the grammatical comprehension task (U = 70.50, *p* < 0.001; d = 1.251).

### 3.6. Analysis of Narrative Production Skills

Data regarding the narrative skills of the two groups of participants are presented in Table 7. Levene’s test for equality of variances showed that the assumption of homogeneity of variance had been violated in both cases (% errors of global coherence: *p* < 0.002; % lexical informativeness: *p* < 0.022). For this reason, potential group-related differences on these two macrolinguistic variables were assessed with two non-parametric Mann–Whitney tests. Statistical significance was set at *p* < 0.025 (0.05/2 dependent variables) after Bonferroni correction for multiple comparisons. These analyses showed the absence of a significant group-related difference in % errors of global coherence (U = 170.00, *p* = 0.557; d = 0.193). However, the participants with DLDs produced significantly fewer lexical information units (U = 100.50, *p* < 0.010; d = 0.871).

### 3.7. Relation between Measures of EF and Linguistic Skills

The potential relation between measures of updating (as measured via the Composite Digit Recall score) and inhibition (as measured via the errors produced on the Part_B (inhibition condition) subtest of the inhibition task of the NEPSY-II) and those linguistic scores on which the participants with DLDs performed significantly lower than controls (i.e., articulation, phonological discrimination, % complete sentences produced on the narrative production task, grammatical comprehension, and % lexical informativeness on the narrative production task) was explored by using the Pearson product-moment correlation coefficient. Namely, for these analyses, we report r-values, *p*-values, and the percentage of variance shared by the two variables (calculated by multiplying by 100 the coefficient of determination, r^2^).

Significant moderate to strong positive correlations were found between scores obtained on the composite digit span recall and measures of articulation (r = 0.411; *p* < 0.008; percentage of shared variance: 17%), phonological discrimination (r = 0.406; *p* < 0.009; percentage of shared variance: 16%), grammatical comprehension (r = 0.613; *p* < 0.001; percentage of shared variance: 38%), and % lexical informativeness (r = 0.318; *p* < 0.045; percentage of shared variance: 10%).

Finally, robust significant negative correlations were found between the production of errors produced on the Part B of the inhibition task and phonological discrimination (r = −0.767; *p* < 0.001; percentage of shared variance: 59%), grammatical comprehension (r = −0.699; *p* < 0.001; percentage of shared variance: 49%), % complete sentences (r = −0.470; *p* < 0.002; percentage of shared variance: 22%), and % lexical informativeness (r = −0.549; *p* < 0.001; percentage of shared variance: 30%).

## 4. Discussion

The current investigation explores executive, linguistic and narrative skills in children with DLDs and TD and the potential relation between measures of updating and inhibition and the abilities to produce and discriminate phonemes, words, and sentences, as well as the ability to organize a narrative discourse at the macrolinguistic level (i.e., global coherence) and utter words that are appropriate from both linguistic and communicative points of view (i.e., lexical informativeness). Overall, the cohort of children with DLDs scored lower than controls on tasks assessing updating (i.e., phonological short-term and working memory) and inhibitory skills (confirmation of Hypothesis 1). As for language, they showed a complex pattern of results with some impaired abilities (e.g., articulation, phonological discrimination, grammatical comprehension and production, and the ability to produce appropriate words on the narrative production task) and other ones where they did not score differently than controls (e.g., lexical production and comprehension skills, and the production of errors of global comprehension while telling a story) (partial confirmation of Hypothesis 2). Finally, updating and inhibitory skills were found related to the difficulties observed in the participants with DLDs (confirmation of Hypothesis 3). These results will be discussed in light of current models of message processing.

From a cognitive point of view, this study confirms previous investigations highlighting in children with DLD potential difficulties in inhibition [18,33,34] and phonological working memory, a cognitive system that plays a crucial role in the updating process (e.g., [17,31,35,36,37]). According to Baddeley [38], working memory allows individuals to momentarily store and process visual and/or verbal information through different components. Namely, verbal information is assumed to be processed in a phonological working memory system formed by a phonological short-term memory system, where incoming verbal information is momentarily stored, and a subvocal rehearsal process that keeps this information active until needed. Participants with DLDs performed worse than controls on both forward and backward digit recall tasks, suggesting a difficulty not only in the passive storage of verbal information in working memory (as investigated via the forward digit recall task) but also in the active manipulation of such information (as investigated via the backward digit recall task). This supports the possibility that reduced phonological working memory skills are a characteristic feature of DLDs. Indeed, a difficulty in keeping track of linguistic information in short-term memory and eventually process it in working memory might result in slowed vocabulary acquisition (e.g., [39,40]) and affect the linguistic development of children with DLDs [35,41]. In line with this hypothesis, phonological short-term and working memory limitations correlated with a range of linguistic measures in the current study. Similarly, the participants with DLDs produced significantly more errors than controls on the two subtests of the inhibition task of the NEPSY-II. It is noteworthy, however, that their performance on such tests was within normal range, as shown by their mean scalar scores (see Table 3). On the naming subtest, 69% of them scored above the 50th centile while none of them scored below the 25th centile; on the inhibition subtest, their performance was more variegated with 44% of them scoring at or above the 50th centile, and 43% of them at or below the 25th centile (8% among children with TD). Altogether, these data suggest a relative weakness in inhibitory control in the participants with DLDs, which is not necessarily pathological. Such variability may be among the causes of the heterogeneous findings reported in the literature on inhibition in children with DLDs, who may present with difficulties in other components of EF (e.g., in [20] where a cohort of twenty-two preschoolers with DLD showed difficulties on both verbal and non-verbal tasks assessing updating and shifting but not on tasks assessing inhibition).

From a linguistic point of view, as a group, the preschoolers with DLDs showed a range of linguistic difficulties affecting both production and comprehension (see also [1]). In line with several previous reports, such difficulties involved mostly phonological and syntactic skills [42,43,44]. A careful evaluation of our results suggests that the children with DLDs did not have articulatory difficulties. Indeed, in the expressive tasks, they did not show articulatory impairments while producing words. On the contrary, their errors were mostly characterized by phonological substitutions (i.e., phonological paraphasias). Together with their significant difficulties in the phonological discrimination task, this rules out the possibility that their difficulties in dealing with the Articulation task were phonetic. Rather they were likely due to a difficulty in phonological processing, i.e., in the ability to categorize phones in abstract phonemic categories. Phonological problems were accompanied by difficulties in both expressive and receptive syntax [45,46] but not by difficulties in naming or lexical comprehension. Indeed, morphology and syntax are usually more affected than vocabulary in these children [47,48]. Their utterances were characterized by reduced grammatical accuracy and omissions of both content and function words that made their sentences incomplete [47,49,50]. Interestingly, even in absence of naming difficulties and more general impairments in the ability to generate a mental model of the story to describe (as shown by the lack of group-related differences in the production of errors of global coherence), the participants with DLDs produced significantly fewer words that were scored as informative during the narrative production task (see also [17]). This is a particularly interesting result as it suggests that, even if they were able to correctly select the target words in a decontextualized setting (as it is the case for naming tasks), they could not efficiently select words within a given narrative context. In other terms, they had difficulties in using contextual elements in the process of lexical selection (see below).

We would also like to stress the heterogeneity of the linguistic performance of the children with DLDs. Indeed, as can be seen in Table 4, Table 5, Table 6 and Table 7, the performance of these participants is quite variable. Even if, as a group, they scored lower than controls on measures assessing phonology (articulation and phonological discrimination), syntax (% complete sentences and grammatical comprehension), and the ability to produce contextually adequate words (% lexical informativeness), in all of these cases, there were individuals scoring up to 2 SDs below the expected mean, as well as children performing 1 or even 2 SDs above the mean. At the same time, significant within-group variations were also found on measures tapping lexical skills (naming and lexical comprehension) with children scoring 2 SDs below the mean. This further confirms the need for a comprehensive assessment of language skills in these children in order to derive a clear picture of their individual relative strengths and weaknesses. These will eventually be necessary for an adequate description of their linguistic profile and for the generation of effective rehabilitation programs.

A final issue regards the need to interpret the linguistic difficulties described so far within a theoretical framework of message production and comprehension. According to some of the most influential models of language processing, the production and comprehension of messages and/or narratives are the result of multistage processes characterized by an interplay between cognitive and linguistic skills (e.g., [22,51]). For example, according to the Structure Building Framework (SBF; [22]), in order to produce a story, a speaker needs to generate a structure or mental depiction of its contents that will serve as a foundation for its development. In this preliminary phase, the ability to focus the attention to the goal at hand and to inhibit potentially distracting actions is crucial. Once the story structure has been generated, the speaker will need to organize it in sequences that must be converted in propositions and eventually verbalized through processes of lexical selection, access, and production (e.g., [21]). During these phases, the possibility to inhibit the generation of irrelevant (e.g., tangential) propositions and the selection of wrong words, as well as the ability to keep all this information active in working memory, are crucial. As the information flows, the speaker also needs to monitor its consistency with the previously generated structure(s) and, in case of inconsistency, shift and generate a new structure. Additionally, message comprehension is likely the result of a multistage process that entails the interaction between bottom-up and top-down processes (e.g., [52]). After decoding the phones uttered by the interlocutor, the listener needs to map the perceived frequencies onto phonemic categories, select the target words, and get access to all of their information that will guide the generation of the meaning of the perceived sentence. The organization of such meanings will eventually trigger the generation of structures and mental depictions of the perceived messages. Furthermore, the comprehension of a message is the result of a complex interplay between cognitive (e.g., EF and attention) and linguistic skills. The cohort of children with DLDs had difficulties in the generation of informative words that, as stated earlier in this discussion, were not the consequence of lexical selection/access difficulties. Indeed, their performance on both naming and lexical comprehension tasks (that directly assess lexical selection/access abilities) was similar to that of the participants with typical linguistic development. It is then likely that the reduced levels of lexical informativeness found in the narratives by preschoolers with DLDs are related to other factors.

Coherently with the above-mentioned models of message production, in our study, the % of lexical informativeness showed a moderate positive correlation with the composite score of digit recall and a large negative correlation with the production errors in the inhibition task. Considering that these two measures of EF shared 10% and 30% of variance with that measures of linguistic efficiency, our findings support the possibility that the ability to select contextually appropriate words during the production of a narrative may be related to the abilities to keep the already generated structures active in working memory (updating) while inhibiting irrelevant and/or inadequate pieces of information. Interestingly, growing experimental evidence suggests that the ability to actively suppress irrelevant, conflicting information that otherwise could cause interference emerges functionally intact in early childhood [53,54,55]. Therefore, our data support the possibility of difficulty in developing a function that should be in place in children aged 4 to 5 years.

The participants with DLDs also had difficulties in later stages of message production. Indeed, even in absence of relevant difficulties of lexical selection, they had significant impairments in the production of complete sentences on the narrative production task and in the articulation of the selected words on the articulation and naming tasks. These findings are coherent with previous investigations (e.g., [47]) and, interestingly, also in these cases, measures of EF showed a significant correlation. Namely, the production of errors on Part B of the inhibition task showed a medium negative correlation with the % of complete sentences. With 22% of shared variance, this correlation suggests that lower inhibitory skills (i.e., the production of more errors on the inhibition task) might significantly affect the process of sentence generation during storytelling. Qualitative inspection of the transcripts confirmed that the majority of the sentences produced by children with DLDs were not complete as they were abruptly interrupted. Usually, these participants searched for the appropriate words that could be inserted in the subsequent utterances. Therefore, reduced efficiency in the ability to suppress irrelevant information might play a role while generating grammatical structures. Finally, a significant medium positive correlation was found between the Composite Digit Span Recall score and the measure of articulation with 17% of shared variance highlighting the possibility of a relation between updating and the ability to correctly articulate words. Nonetheless, significant correlations were also found between measures of updating and inhibition and measures of linguistic comprehension. The ability to identify minimal pairs had a moderate positive correlation with the Composite Digit Span Recall score with 16% of shared variance and a strong negative correlation with the production of errors in Part B of the inhibition task. In this case, the two variables shared 59% of the variance. Therefore, the process of phonological discrimination is also likely affected by EF. In the same vein, grammatical comprehension showed a significant positive correlation with the Composite Digit Span Recall score, with 38% of shared variance and a strong negative correlation with the production of errors on Part B of the inhibition task with 49% of shared variance. Overall, then, these correlations highlight the complexity of linguistic development and processing in children with DLDs.

## 5. Conclusions

In conclusion, these findings suggest that DLD is a complex developmental disorder whose symptoms are linked to the interaction among several factors. The current investigation supports previous reports about the presence of potential difficulties of updating and inhibition in children with DLDs. Furthermore, it also suggests that such difficulties, even if mild, may be related to their linguistic abilities. One of the limitations of the current investigation concerns the small number of participants that did not allow us to perform more accurate analyses (e.g., regression models) to explore the nature of the relationship between measures of updating and inhibition and linguistic and narrative ones. Indeed, in line with the current literature, even if we have discussed a potential causal relation between EF and linguistic functions, the reverse relationship may also be possible, with linguistic difficulties impacting the ability to adequately perform on tasks assessing executive functions. Of note, in our study, the tasks assessing updating and inhibition were inherently verbal. This further highlights the need to also use non-verbal tasks to assess these functions in order to rule out a potential linguistic bias. Future studies should replicate these findings on larger cohorts of participants in order to explore the potential causative relation between difficulties in EF and linguistic/narrative skills. Nonetheless, the current results suggest the need to include the assessment of EF in preschoolers with DLDs and to embed the rehabilitation of such functions in the early intervention of these children.

## Figures and Tables

**Table 1 ijerph-17-01772-t001:** Age, gender, and performance on a non-verbal reasoning task (i.e., Raven’s matrices) of the two groups of participants. Data are presented as means (and standard deviations, SD) for age, number of participants, and percentage for gender and raw scores (mean and SD) for Raven’s matrices.

General Information	TD	DLD
	(*N* = 24)	(*N* = 16)
Age	5.43 (0.46)	5.19 (0.03)
Gender	M = 15 (63%)	M = 12 (75%)
Raven’s Matrices	22.33 (4.40)	18.75 (4.75)

Note: TD = children with typical development; DLD = children with developmental language disorders.

**Table 2 ijerph-17-01772-t002:** Results from the assessment of updating skills. Data are presented as means (and standard deviations, SD). The asterisk (*) shows when the group-related difference was significant.

Assessment of Updating Skills	TD	DLD
Forward Digit Recall *	6.00 (1.25)	4.75 (1.73)
Backward Digit Recall *	3.04 (0.95)	2.00 (1.10)
Composite Digit Recall *	9.04 (1.71)	6.75 (2.15)

Note: TD = children with typical development; DLD = children with developmental language disorders. *:The asterisk signals when a group-related difference is significant.

**Table 3 ijerph-17-01772-t003:** Scalar scores obtained by the two groups of participants on the inhibition test of the NEPSY-II (Parts A and B, respectively). The number of participants having a specific scalar score for each group is presented together with percentages.

**Inhibition A (errors)**	**TD**	**DLD**
>75	16 (67%)	5 (31%)
51–75	2 (8%)	6 (38%)
26–50	4 (17%)	0 (--)
11–25	2 (8%)	5 (31%)
6–10	0 (--)	0 (--)
2–5	0 (--)	0 (--)
**Inhibition B (errors)**		
>75	19 (79%)	3 (19%)
51–75	2 (8%)	4 (25%)
26–50	1 (4%)	2 (13%)
11–25	1 (4%)	5 (31%)
6–10	1 (4%)	1 (6%)
2–5	0 (--)	1 (6%)

Note: TD = children with typical development; DLD = children with developmental language disorders.

**Table 4 ijerph-17-01772-t004:** Results (means and SDs) of the assessment of articulation and phonological discrimination skills in the two groups of participants. For children with DLDs, both raw scores and z-scores with respect to normative data are reported in separate columns. For z-scores, ranges are also reported. The asterisk (*) shows when the group-related difference in raw scores is significant.

Assessment of Articulation and Phonological Discrimination	TD	DLD	DLD
	(Raw Score)	(Raw Score)	(z-Score)
Articulation *	143.92 (5.91)	126.06 (20.29)	−0.91 (1.02) – Range: -2/1
Phonological Discrimination *	97.71 (4.66)	80.94 (24.65)	−0.47 (1.18) – Range: -2/2

Note: DLD = children with developmental language disorders; TD = children with typical language development. *: The asterisk signals when a group-related difference is significant.

**Table 5 ijerph-17-01772-t005:** Results (means and SDs) of the assessment of naming and lexical comprehension skills in the two groups of participants. For children with DLDs, both raw scores and z-scores with respect to normative data are reported in separate columns. For z-scores, ranges are also reported. The asterisk (*) shows when the group-related difference in raw scores is significant.

Assessment of Lexical Skills	TD	DLD	DLD
	(Raw Score)	(Raw Score)	(z-Score)
Naming	69.00 (4.75)	66.06 (6.08)	0.06 (1.17) – Range: -2/2
Lexical Comprehension	16.04 (1.43)	15.38 (1.82)	−0.63 (0.87) – Range: -2/1

Note: DLD = children with developmental language disorders; TD = children with typical language development.

**Table 6 ijerph-17-01772-t006:** Results (means and SDs) of the assessment of grammatical production and comprehension skills in the two groups of participants. For children with DLDs, both raw scores and z-scores with respect to normative data are reported in separate columns. For z-scores, ranges are also reported. The asterisk (*) shows when the group-related difference in raw scores is significant.

Assessment of Grammatical Skills	TD	DLD	DLD
	(Raw Score)	(Raw Score)	(z-Score)
% Complete Sentences *	52.33 (13.10)	41.00 (14.20)	−0.47 (0.65) – Range: -1.5/0
Grammatical Comprehension *	34.00 (4.05)	26.13 (8.96)	−0.69 (0.93) – Range: -2/1

Note: DLD = children with developmental language disorders; TD = children with typical language development. *: The asterisk signals when a group-related difference is significant.

**Table 7 ijerph-17-01772-t007:** Results (means and SDs) of the assessment of narrative skills in the two groups of participants. For children with DLDs, both raw scores and z-scores with respect to normative data are reported in separate columns. For z-scores also ranges are reported. The asterisk (*) shows when the group-related difference in raw scores is significant.

Assessment of Narrative Skills	TD	DLD	DLD
	(Raw Score)	(Raw Score)	(z-score)
% Errors of Global Coherence	4.89 (6.07)	9.42 (13.14)	0.22 (.60) – Range: 0 / 2
% Lexical Informativeness *	86.19 (7.07)	76.77 (13.46)	−0.13 (.70) – Range: -1.5/1

Note: DLD = children with developmental language disorders; TD = children with typical language development. *: The asterisk signals when a group-related difference is significant.

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
