# Peer review of "The Complex Relation between Executive Functions and Language in Preschoolers with Developmental Language Disorders"

_ijerph, 2020, doi:10.3390/ijerph17051772_

Round 1

Reviewer 1 Report

The research described in the manuscript compared children with developmental language disorders (DLD) and typically developing children on a range of executive function (EF)  and linguistic tasks. Results show that children with DLD performed worse on most of the measures, and furthermore, there was a correlation between EF and the linguistic measures, although this correlation varied depending on the measured linguistic function.

The major limitation of the research, which the authors themselves mention, is the small number of participants, especially in the DLD group.  This affects not only the statistical analysis that can be performed, but also the interpretation of specific differences as representative and generalizable. Despite this, I think the overall data are interesting.

The relation between EF and linguistic skills: I think it would be helpful if the authors explain the variation in terms of significance and strength of the relation between EF and different linguistic functions.

The paragraph the begins on Line 74 should be written. As it is written now it is unclear, and it's hard to understand what was found. If the issue is whether impairments were found on non-verbal tasks, the authors should clearly present this issue, then present the relevant finding, first those supporting the existence of such impairments than those that did not find such impairments. The authors begin with studies that showed general attention difficulties, then mention a study that did not find difficulties  on non-verbal tasks, then report a meta-analysis that showed worse performance on non-verbal tasks, then refer to studies showing problems with verbal tasks, then refer to a study that didn’t find problems on a non-verbal task. This is very difficult to follow.

Although the authors imply that the potential causal relation between EF and linguistic functions is such that EF impairment lead to linguistic impairments, I think it should be pointed out that the reverse relation is also possible, for example given findings on the bilingual advantage in executive function, and should be discussed.

Statistical analyses comments

I think the following points should be explained addressed in the paper

  • What is the purpose of analyzing a composite digit recall score in addition to the digit forward and digit backward measures?
  • Why were both X-scores and scalar scores on the inhibition test used?
  • The between group differences on the Raven matrices are marginally significant, given the small number of participants it could well be a power issue.

Minor comments

  • Participation in the study not to the study (Line 126)
  • Does "infancy school" mean preschool?
  • I think you meant to say results are inconsistent, rather than controversial (Line 83)
  • I am not sure what "or an alternate response" refers to (Line 149)

Author Response

Dear Reviewer 1,

on behalf of all the authors thank you for the comments and suggestions made to our paper. Attached you can find the answers to the questions you raised.

Best regards,

livia taverna

Reviewer 2 Report

Please see in the attachment

Author Response

Dear Reviewer 2,

on behalf of all the authors thank you for the comments and suggestions made to our paper. Attached you can find the answers to the questions you raised.

Best regards,

livia taverna

Round 2

Reviewer 2 Report

The authors have adequately addressed all of my concerns.